# Other Topics You May Also Agree or Disagree:
# Modeling Inter-Topic Preferences using Tweets and Matrix Factorization

## Abstract

This paper presents an approach for modeling inter-topic preferences of Twitter users: for example, *those who agree TPP also agree free trade*. This kind of knowledge is useful not only for stance detection across multiple topics but also for various real-world applications including public opinion survey, election prediction, election campaign, and online debate. In order to extract users' preferences on Twitter, we design high-quality linguistic patterns (e.g., "$A$ is completely wrong") in which people agree and disagree topics. By applying the linguistic patterns to a collection of tweets, we extract statements agreeing and disagreeing various topics. Inspired by the work on item recommendation, we formalize the task of modeling inter-topic preferences as matrix factorization: representing users' preference as a user-topic matrix and mapping both users and topics into a latent feature space that abstracts the preferences. The experimental results demonstrate that the presented approach is useful for predicting missing preferences of users and that the latent vector representations of topics encode inter-topic preferences.

## 1 Introduction

Social media have changed the way people shape public opinions. The latest survey by Pew Research Center reported that a majority of US adults (62%) obtain news on social media, and 18% do so often (Gottfried and Shearer, 2016). As news and opinions are shared and amplified by friend networks of individuals (Jamieson and Cappella, 2008), individuals are also isolated from information that does not fit their opinions (Pariser, 2011). Ironically, the cutting-edge technology of social media promotes ideological groups even with its potential to deliver diverse information.

A great deal of studies analyze discussions, interactions, influences, and communities on social media along the political spectrum of liberal to conservative (Adamic and Glance, 2005; Zhou et al., 2011; Cohen and Ruths, 2013; Bakshy et al., 2015; Wong et al., 2016). Even though these studies provide intuitive visualizations and interpretations along the axis of liberal-conservative, political analysts argue that the axis is flawed and insufficient for representing public opinions and ideologies (Kerlinger, 1984; Maddox and Lilie, 1984).

A potential solution for analyzing multiple axes of political spectrum on social media is stance detection (Thomas et al., 2006; Somasundaran and Wiebe, 2009; Murakami and Raymond, 2010; Anand et al., 2011; Walker et al., 2012; Mohammad et al., 2016; Johnson and Goldwasser, 2016), whose task is to determine whether the author of a text is for, neutral, or against towards a topic (e.g., *free trade*, *immigration*, *abortion*). However, stance detection across different topics is extremely difficult. Anand et al. (2011) reported that a sophisticated method with topic-dependent features greatly improved the performance of stance detection within a topic, but could not outperform a baseline method with simple $n$-gram features when evaluated across topics. More recently, all participants of SemEval 2016 Task 6A (with five topics) could not outperform the baseline supervised method using $n$-gram features (Mohammad et al., 2016).

In addition, stance detection encounters the difficulty with different user types. Cohen and Ruths (2013) observed that existing methods on stance detection fail on 'ordinary' users because they mostly obtain training and test data from polit-

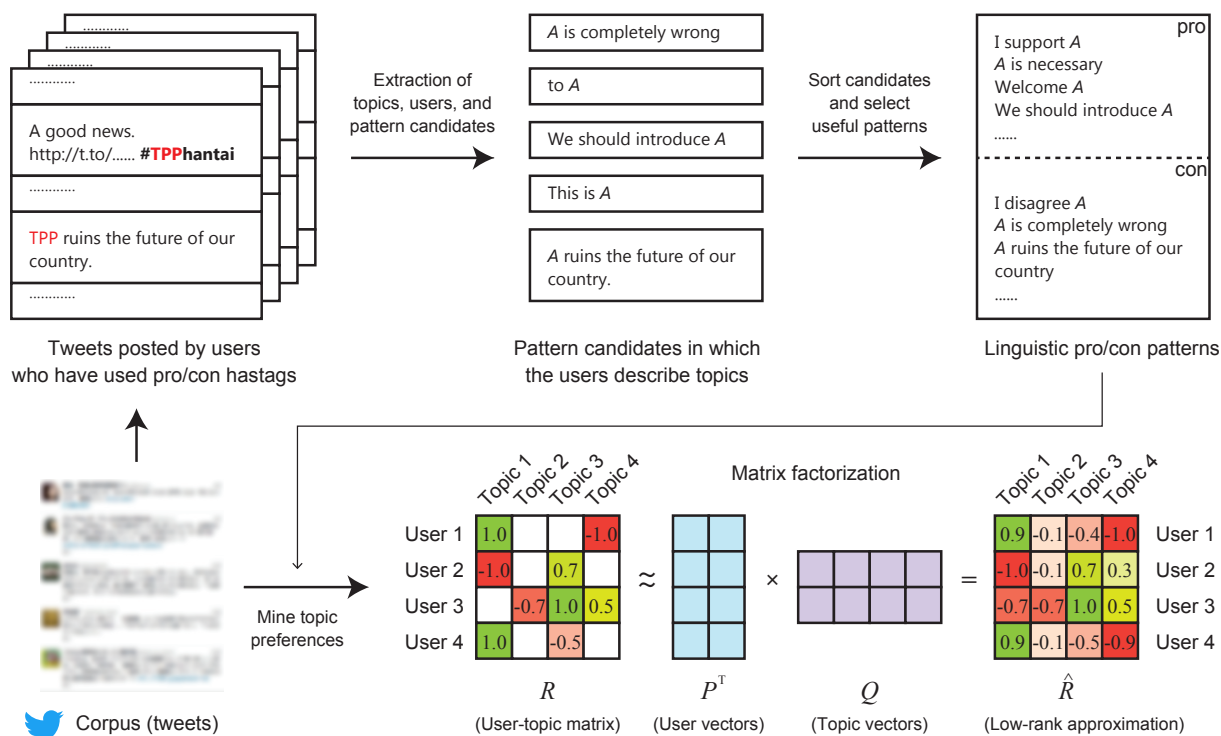

Figure 1: A generic overview of this study.

ically vocal users (e.g., politicians); for example, they found that a stance detector trained on a dataset with politicians achieved 91% accuracy on other politicians but only 54% accuracy on 'ordinary' users. Establishing a bridge across different topics and users is a major challenge not only to stance detection but also to social media analytics.

An important material for the bridge is commonsense knowledge about topics. For instance, consider a topic *a revision of Article 96 of the Japanese Constitution*. We infer that a statement, "we should maintain armed forces," tends to favor the topic even without any lexical overlap between the topic and statement. This inference is reasonable because: the writer of the statement favors *armed forces*; those who favor *armed forces* also favor *a revision of Article 9*[1]; and those who favor *a revision of Article 9* also favor *a revision of Article 96*[2]. In general, this kind of commonsense knowledge can be expressed in the format: *those who agree/disagree a topic A also agree/disagree another topic B*. We call this kind of knowledge *inter-topic preference* throughout this paper.

We conjecture that the previous work on stance detection indirectly learns inter-topic preferences

---

[1] Article 9 prohibits armed forces in Japan.

[2] Article 96 specifies high requirements for making amendments to Constitution of Japan (including Article 9).

within the same target through the use of $n$-gram features on a supervision data. In contrast, this paper directly acquires inter-topic preferences from an unlabeled corpus of tweets. The knowledge of inter-topic preferences is useful not only for stance detection but also for various real-world applications including public opinion survey, election campaign/prediction, and online debate.

Figure 1 illustrates a generic overview of this work. We extract high-quality linguistic patterns (e.g., "$\underline{A}$ is completely wrong") in which people agree and disagree topics, making use of hashtags in a large collection of tweets (Section 2.1). The patterns are used for extracting instances of users' preferences to various topics (Section 2.2). Inspired by the work on item recommendation, Section 3 formalizes the task of modeling inter-topic preferences as a matrix factorization: representing a sparse user-topic matrix (the extracted instances) with the product of low-rank user and topic matrices. The low-rank matrices provide *latent vector representations* of users and topics. This approach is also useful for completing preferences of 'ordinary' (less vocal) users, which fills the gap between different types of users.

The contributions of this paper are three-folds.

1. To the best of our knowledge, this is the first

study that models inter-topic preferences for unlimited targets on the real-world data.

2. The experimental results show that this approach can predict missing topic preferences of users accurately (80–94%).

3. The experimental results also demonstrate that the latent vector representations of topics encode inter-topic preferences, e.g., *those who agree nuclear power plant also agree nuclear fuel cycle*.

This study uses a Japanese Twitter corpus because of its availability from the authors, but the core idea is applicable to any language.

## 2 Mining topic preferences of users

This section collects statements of users agreeing or disagreeing various topics on Twitter as source data for modeling inter-topic preferences. More formally, we are interested in acquiring a collection of tuples $(u, t, v)$, where: $u \in U$ is a user; $U$ is the set of all users on Twitter; $t \in T$ is a topic; $T$ is the set of all topics; and $v \in \{+1, -1\}$ is $+1$ when the user $u$ agree the topic $t$ and $-1$ otherwise (disagreement).

Throughout this work, we use a corpus consisting of 35,328,745,115 Japanese tweets (7,340,730 users) crawled from February 6, 2013 to September 30, 2016. We removed retweets from the corpus.

### 2.1 Mining linguistic patterns of agreement and disagreement

We use linguistic patterns to extract tuples $(u, t, v)$ from the corpus. More specifically, when a tweet message matches to one of linguistic patterns for agreement (e.g., "$\underline{t}$ is necessary"), we regard that the author $u$ of the tweet agrees the topic $t$. Conversely, a statement of disagreement is identified by linguistic patterns for disagreement (e.g., "$\underline{t}$ is unacceptable").

In order to design high-quality linguistic patterns, this study focuses on hashtags appearing in the corpus, which have been popular clues for locating subjective statements such as sentiments (Davidov et al., 2010), emotions (Qadir and Riloff, 2014), and ironies (Van Hee et al., 2016). Hashtags are also useful for finding strong supporters and critics and their target topics; for example, #immigrantsWelcome in-

dicates that the author favors *immigrants*; and #StopAbortion is against *abortion*.

Based on this intuition, we design regular expressions for *pro hashtags* "#(.+)sansei"[3] and for *con hashtags* "#(.+)hantai"[4], where (.+) matches to a target topic. These regular expressions can find users who have strong preferences to topics. In this way, we could extract 31,068 occurrences of pro/con hashtags, which were used by 18,582 users for 4,899 topics. We regard the topics found in this procedure as the set of target topics $T$ in this study.

Every time we find a tweet containing a pro/con hashtag, we look for corresponding textual statements as follows. Suppose that a tweet mentions a hashtag (e.g., #TPPsansei) for a topic $t$ (e.g., *TPP*). Assuming that the author of the tweet does not change their attitude to a topic over time, we search for other tweets posted by the same author with the topic keyword $t$. This process retrieves tweets like "I support TPP." Then, we replace the topic keyword into a variable $A$ to extract patterns, e.g., "I support $\underline{A}$." Here, the definition of the pattern unit is language specific. For Japanese tweets, we simply recognize a pattern starting at a variable (topic) and ending at the end of the sentence[5].

Because this procedure also extracts unuseful patterns such as "to $A$" and "this is $A$", we manually choose useful patterns in a systematic way: sort patterns in descending order of the number of users who use a pattern; and check the sorted list of patterns manually; and remove unuseful patterns. In this way, we obtain 100 pro patterns (e.g., "welcome $A$" and "$A$ is necessary") and 100 con patterns ("do not let $A$" and "I don't want $A$").

### 2.2 Extracting instances of topic preferences

By using the pro and con patterns acquired in Section 2.1, we extract instances of $(u, t, v)$ as follows. When a sentence in a tweet (whose author is the user $u$) matches to one of pro patterns (e.g., "$\underline{t}$ is necessary") and the topic $t$ is included in the set of target topics $T$, we recognize this as an instance

---

[3] Unlike English hashtags, we systematically attach a noun *sansei*, which stands for *pro* (agreement) in Japanese, to a topic, for example, #TPPsansei. This paper uses the alphabetical expression sansei only for explanation; the actual pattern uses Chinese characters corresponding to *sansei*.

[4] A Japanese noun *hantai* stands for *con* (disagreement), for example, #TPPhantai. This paper uses the alphabetical expression hantai only for explanation; the actual pattern uses Chinese characters corresponding to *hantai*.

[5] In English, this treatment roughly corresponds to extracting a verb phrase with the variable $A$.

of $(u, t, +1)$. Similarly, when a sentence matches to one of con patterns (e.g., "I don't want $\underline{t}$") and the topic $t$ is included in the set of target topics $T$, we recognize this as an instance of $(u, t, -1)$. In this way, we collected 25,805,909 tuples regarding 3,302,613 users and 4,899 topics. Because these collected tuples also including far less frequent users and topics, we remove users and topics appeared less than 5 times. In addition, there also meaningless frequent topics such as "of" and "it". Therefore, we sorted topics in descending order of co-occurrence frequency with each of pro patterns and con patterns, and remove meaningless topics in the top 100 topics. This resulted in 9,961,509 tuples regarding 273,417 users and 2,323 topics.

## 3   Matrix factorization

Section 2 collects a number of instances of users' preferences to various topics from the corpus. However, a Twitter user does not necessarily express preferences for all topics. In addition, it is by nature impossible to predict whether a new (non-existent in the data) user agree or disagree topics. Therefore, this section applies matrix factorization (Koren et al., 2009) in order to predict missing values, inspired by the work of item recommendation (Bell and Koren, 2007; Dror et al., 2011). In essence, matrix factorization maps both users and topics into a latent feature space that abstracts topic preferences of users.

Let $R$ be a sparse matrix of $|U| \times |T|$. Only when a user $u$ expresses a preference to topic $t$, we compute an element of the sparse matrix $r_{u,t}$,

$$r_{u,t} = \frac{\#(u, t, +1) - \#(u, t, -1)}{\#(u, t, +1) + \#(u, t, -1)} \quad (1)$$

Here, $\#(u, t, +1)$ and $\#(u, t, -1)$ are the numbers of occurrences of instances $(u, t, +1)$ and $(u, t, -1)$, respectively. Thus, an element $r_{u,t}$ approaches $+1$ as the user $u$ favors the topic $t$, and $-1$ otherwise. If the user $u$ does not make any statement about the topic $t$, i.e., neither $(u, t, +1)$ nor $(u, t, -1)$ exists in the data, we do not fill the corresponding element, leaving it as a missing value.

Matrix factorization decomposes the sparse matrix $R$ into low dimensional matrices $P \in \mathbb{R}^{k \times |U|}$ and $Q \in \mathbb{R}^{k \times |T|}$, where $k$ is a parameter to specify the number of dimension of the latent space. We minimize the following objective function for

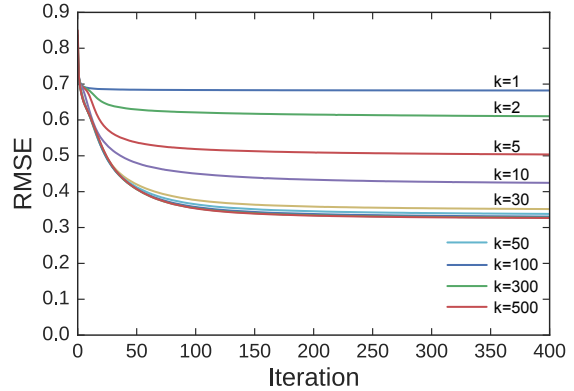

Figure 2: Reconstruction error (RMSE) of matrix factorization with different $k$.

finding the matrices $P$ and $Q$,

$$\min_{P,Q} \sum_{(u,t) \in R} \left( (r_{u,t} - \boldsymbol{p}_u^\mathsf{T} \boldsymbol{q}_t)^2 \right.$$
$$\left. + \lambda_P \|\boldsymbol{p}_u\|^2 + \lambda_Q \|\boldsymbol{q}_t\|^2 \right). \quad (2)$$

Here, $(u, t) \in R$ is repeated for elements filled in the sparse matrix $R$, $\boldsymbol{p}_u \in \mathbb{R}^k$ and $\boldsymbol{q}_v \in \mathbb{R}^k$ are $u$ column vectors of $P$, and $v$ column vectors of $Q$, respectively, and $\lambda_P \geq 0$ and $\lambda_Q \geq 0$ present coefficients of regularization terms. We call $\boldsymbol{p}_u$ and $\boldsymbol{q}_t$ the *user vector* and *topic vector*, respectively.

Using the user and topic vectors, we can predict an element $\hat{r}_{u,t}$ that may be missing in the original matrix $R$,

$$\hat{r}_{u,t} \simeq \boldsymbol{p}_u^\mathsf{T} \boldsymbol{q}_t. \quad (3)$$

We use `libmf`[6] (Chin et al., 2015) for solving the optimization problem in Equation 2. We set the regularization coefficients $\lambda_P = 0.1$ and $\lambda_Q = 0.1$, and use default values for other parameters of `libmf`.

## 4   Evaluation

### 4.1   Determining the dimension parameter $k$

How good is the low-rank approximation found by matrix factorization? What is the sweet spot for the number of dimension $k$ of the latent space? We investigate the reconstruction error of matrix factorization with different values of $k$ in order to answer these questions. We use Root Mean Squared

---

[6] https://github.com/cjlin1/libmf

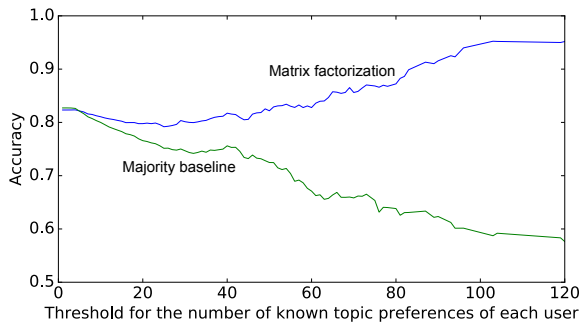

Figure 3: Prediction accuracy when changing the threshold for the number of known topic preferences of each user.

Error (RMSE) for measuring the error,

$$RMSE = \sqrt{\frac{\sum_{(u,t)\in R}(\boldsymbol{p}_u^\mathsf{T}\boldsymbol{q}_t - r_{u,t})^2}{N}}. \quad (4)$$

Here, $N$ is the number of elements in the sparse matrix $R$ (the number of known values).

Figure 2 shows RMSE values over iterations of `libmf` with the dimension parameter $k \in \{1, 2, 5, 10, 30, 50, 100, 300, 500\}$. We can observe that a reconstruction error decreases as the iterative method of `libmf` progresses. The larger the number of dimension $k$ is, the less the reconstruction error becomes; the lowest reconstruction error was 0.3256 with $k = 500$. It is also interesting to observe the error with $k = 1$, which corresponds to mapping users and topics onto one dimension similarly to the political spectrum of liberal and conservative. Judging from the relatively high RMSE values with $k = 1$, it may be difficult to represent everything in the data on a one-dimensional axis. Based on this result, we draw a conclusion that matrix factorization with $k = 100$ is sufficient to reconstruct the original matrix $R$, and use this parameter value in the rest of the experiments.

## 4.2 Predicting missing topic preferences

How accurately can the user and topic vectors predict missing topic preferences? In order to investigate this question, we evaluate the accuracy for predicting hidden preferences in the matrix $R$ as follows. First, we randomly select 5% of existing elements in $R$, and let $Y$ the collection of the selected elements (test set). We then perform matrix factorization on the sparse matrix without the selected elements of $Y$ and only with the remaining

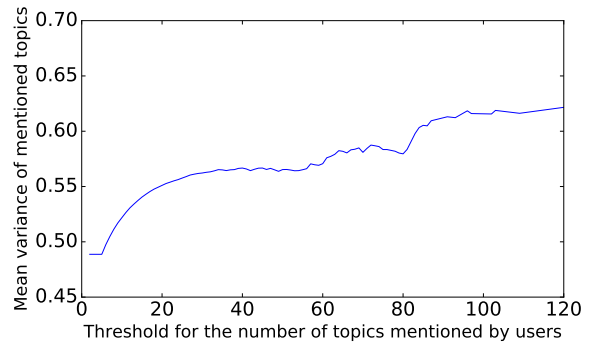

Figure 4: Mean variance of preference values of self-declared topics when changing the threshold for the number of self-declared topics.

95% elements of $R$ (training set). The accuracy of the prediction is defined by,

$$\frac{1}{|Y|} \sum_{u,t\in Y} \mathbb{1}\left(\text{sign}(\hat{r}_{u,t}) = \text{sign}(r_{u,t})\right) \quad (5)$$

Here, $r_{u,t}$ denotes the actual (self-declared) preference values, $\hat{r}_{u,t}$ is the preference value predicted by Equation 3, $\text{sign}(.)$ presents the sign of the argument, and $\mathbb{1}(.)$ yields 1 only when the condition described in the argument holds and 0 otherwise. In other words, Equation 5 computes the proportion of correct predictions, assuming zero to be the decision boundary between pro and con.

Figure 3 plots prediction accuracy values calculated from different sets of users. Here, the x-axis presents a threshold $\theta$ that filters out users whose declarations of topic preferences are no greater than $\theta$ topics. In other words, Figure 3 shows prediction accuracy when we know users' preferences for at least $\theta$ topics. For comparison, we also include the majority baseline that predicts pro and con based on the majority of preferences about each topic in the training set.

The presented method could predict missing preferences with 82.1% accuracy for users stating preferences for at least 5 topics. The accuracy increases as the method receives more information about the users, reaching 94.0% accuracy when $\theta = 100$. This result again indicates that the presented method reasonably utilize known preferences to complete missing preferences.

In contrast, the majority baseline decreases its performance as it receives more information about the users. Because this result was counterintuitive, we examined the cause of this phenomenon. Consequently, this result turned out reasonable because preferences of vocal users deviate

from those of the average users. Figure 4 illustrates this finding, showing the mean of variances of preference values $r_{u,t}$ of all topics. The x-axis presents a threshold $\theta$ that filters out users whose statements of topic preferences are no greater than $\theta$ topics. We can observe that the mean variance increases as we focus on vocal users. These results demonstrate the usefulness of user and topic vectors for predicting missing preferences.

Table 1 shows examples where missing preferences of two users are predicted from known statements of agreements and disagreements[7]. A predicted topic accompanies with the value of $\hat{r}_{u,t}$ in parentheses. For example, the proposed method predicts that the user A, who is positive to *regime change* but negative to *Okinawa US military base*, may also be positive to *vote of non-confidence to Cabinet* but negative to *construction of a new base*.

### 4.3 Inter-topic preferences

Does the topic vectors obtained by matrix factorization capture inter-topic preferences such as "People who agree with A also agree with B"?

Because no dataset exists for this evaluation, we created a dataset of pairwise inter-topic preferences by using a crowdsourcing service[8]. Sampling topic pairs randomly, we collected 150 topic pairs whose cosine similarities of topic vectors are below $-0.6$, 150 pairs whose cosine similarities are between $-0.6$ and $0.6$, and 150 pairs whose cosine similarities are above $0.6$. In this way, we obtained 450 topic pairs for evaluation.

Given a pair of topics $A$ and $B$, a crowd worker is asked to choose a label from: (a) *those who agree/disagree a topic A may also agree/disagree topic B*; (b) *those who agree/disagree a topic A may conversely disagree/agree topic B*; (c) *otherwise (no association between A and B)*. Creating twenty pairs of topics as a gold data, we removed labeling results from workers whose accuracy is lower than 90%.

Consequently, we obtained 6–10 human judgements for every topic pair. Regarding (a) as $+1$ point, (b) as $-1$ point, and (c) as 0 point, we com-

pute the mean of the points (as the average human judgements) for every topic pair. Spearman's rank correlation coefficient ($\rho$) between cosine similarity values of topic vectors and human judgements was 0.2210. We could observe a moderate correlation even though inter-topic preferences collected in this manner are highly subjective.

In addition to the quantitative evaluation, we also check similar topics for three controversial topics, *Liberal Democratic Party (LDP)*, *constitutional amendment* and *right of foreigners to vote* (Table 2). The similar topics to *Liberal Democratic Party (LDP)* were synonymous ones (e.g., *Abe's LDP*, *Abe administration*) as well as other topics promoted by the *LDP* (e.g., *resuming nuclear power plant operations*, *bus rapid transit (BRT)*, *hate speech countermeasure law*). Considering that people who support the LDP may also tend to favor its policies, we found these results reasonable. In the other example, *constitutional amendment* has a similar feature vector to *amendment of Article 9*, *enforcement of specific secrete protection law* and *security related law*. From these results, we conclude that topic vectors were able to capture inter-topic preferences.

## 5 Related work

This section summarizes the related work that spreads across various research fields.

**Social science and political science** A number of of studies analyze social phenomena regarding political activities, political thoughts, and public opinions on social media. These studies model the political spectrum of liberal to conservative (Adamic and Glance, 2005; Zhou et al., 2011; Cohen and Ruths, 2013; Bakshy et al., 2015; Wong et al., 2016), political parties (Tumasjan et al., 2010; Boutet et al., 2013; Makazhanov and Rafiei, 2013), and elections (O'Connor et al., 2010; Conover et al., 2011).

Employing a single axis (e.g., liberal to conservative) or a few axes (e.g., political parties and candidates of elections), these studies provide intuitive visualizations and interpretations along the axes. In contrast, this study is the first attempt for recognizing and organizing various axes of topics on social media with no prior assumption about axes. Therefore, we think this study provides a new tool for computational social science and political science to analyze and interpret phenomena on social media.

---

[7]We anonymized user names in these examples. In addition, we removed topics that are too discriminatory or aggressive to other countries and races. Even though the experimental results of this paper do not necessarily reflect our idea, we do not think it is a good idea to distribute politically incorrect ideas through this paper.

[8]We used Yahoo! Crowdsourcing, a Japanese online service for crowdsourcing. http://crowdsourcing.yahoo.co.jp/

| User | Type | Topic |
|---|---|---|
| A | Agreement (declared) | regime change, capital relocation |
| | Disagreement (declared) | Okinawa US military base, nuclear weapons, TPP, Abe Cabinet, Abe government, nuclear cycle, right to collective defense, nuclear power plant, Abenomics |
| | Agreement (predicted) | same-sex partnership ordinance (0.9697), vote of non-confidence to Cabinet (0.9248), national people's government (0.9157), abolition of tax (0.8978) |
| | Disagreement (predicted) | steamrollering war bill (-1.0522), worsening dispatch law (-1.0301), Sendai nuclear power plant (-1.0269), war bill (-1.0190), constructing new base (-1.0186), Abe administration (-1.0173), landfill Henoko (-1.0158), unreasonable arrest (-1.0113) |
| B | Agreement (declared) | visit shrine, marriage |
| | Disagreement(declared) | tax increase, conscription, amend Article 9 |
| | Agreement (predicted) | national people's government (0.8467), abolition of tax (0.8300), same-sex partnership ordinance (0.7700), security bills (0.6736) |
| | Disagreement (predicted) | corporate tax cuts (-1.0439), Liberal Democratic Party's draft constitution (-1.0396), radioactivity (-1.0276), rubble (-1.0159), nuclear cycle (-1.0143) |

Table 1: Examples of agreement/disagreement topics predicted for each user (predicted score $\hat{r}_{u,v}$ is shown in parenthesis).

| Topic | Topics with a high degree of cosine similarity |
|---|---|
| Liberal Democratic Party (LDP) | Abe's LDP (0.3937), resuming nuclear power plant operations (0.3765), bus rapid transit (BRT) (0.3410), hate speech countermeasure law (0.3373), Henoko relocation (0.3353), C-130 (0.3338), Abe administration (0.3248), LDP & Komeito (0.2898), Prime Minister Abe (0.2835) |
| constitutional amendment | amendment of Article 9 (0.4520), enforcement of specific secrete protection law (0.4399), security related law (0.4242), specific confidentiality protection law (0.4022), security bill amendment (0.3977), defense forces (0.3962), my number law (0.3874), collective self-defense rights (0.3687), militarist revival (0.3567) |
| right of foreigners to vote | human rights law (0.5405), anti-discrimination law (0.5376), hate speech countermeasure law (0.5080), foreigner's life protection (0.4553), immigration refugee (0.4520), co-organized Olympics (0.4379) |

Table 2: Similar topics for three controversial topics.

Here, we mention some work that acquires lexical knowledge about politics. Sim et al. (2013) measured ideological positions of candidates of US Presidential elections from their speeches. The study first constructs "cue lexicons" from political writings labeled with ideologies by domain experts, using sparse additive generative models (Eisenstein et al., 2011). The constructed cue lexicons are associated with ideologies such as *left*, *center*, and *right*. Representing each speech of a candidate with cue lexicons, they infer the proportions of ideologies of the candidate. The study requires a predefined set of labels and text data associated with the labels.

Bamman and Smith (2015) presented an unsupervised method for assessing the political stance of a proposition like "global warminig is a hoax," along the political spectrum of liberal to conservative. In their work, a proposition is represented by a tuple in the form of ⟨subject, predicate⟩, e.g., ⟨*global warminig*, *hoax*⟩. They presented a generative model for users, subjects, and predicates to find a one-dimensional latent space, which corresponds to the political spectrum.

Similarly to ours, their work (Bamman and Smith, 2015) does not requires labeled data for mapping users and topics (subjects) into a latent feature space. Their paper reported that the generative model outperformed Principal Component Analysis (PCA), which is a method for matrix factorization. The empirical result probably reflected the underlying assumptions that PCA treats missing elements as zero (not as simply missing data). In contrast, this work properly distinguishes missing values from zero, excluding missing elements of the original matrix from the objective function of Equation 2. Furthermore, this work demonstrated the usefulness of the latent space, i.e., topic and user vectors, for predicting missing topic preferences of users and inter-topic preferences.

**Fine-grained opinion analysis** The method presented in Section 2 is an instance of fine-grained opinion analysis (Wiebe et al., 2005; Choi et al., 2006; Johansson and Moschitti, 2010; Yang and Cardie, 2013; Deng and Wiebe, 2015), which extracts a tuple of a subjective opinion, a holder of

the opinion, and a target of the opinion from text. Although these previous studies have a potential to improve the quality of the user-topic matrix $R$, unfortunately, no corpus nor resource is available for Japanese language. We do not have a large collection of English tweets at this moment, but combining fine-grained opinion analysis with matrix factorization is an immediate future work.

**Causality relation**  Some of inter-topic preferences in this work can be explained by causality relation, for example, "TPP promotes free trade." A number of previous studies acquire instances of causal relation (Girju, 2003; Do et al., 2011) and promote/supress relation (Hashimoto et al., 2012; Fluck et al., 2015) from text. The causality knowledge is useful for predicting (hypotheses of) future events (Radinsky et al., 2012; Radinsky and Davidovich, 2012; Hashimoto et al., 2015).

However, inter-topic preferences also include pairs of topics where causality relation hardly holds. For example, it is unreasonable to infer that *nuclear plant* and *railroading of bills* have causal relation, but those who dislike *nuclear plant* also oppose *railroading of bills* because (they think) the governing political parties rush the bill for resuming a nuclear plant. This study models this kind of inter-topic preferences based on the preferences of the public. Having said that, it is a promising future direction of this work to incorporate the approach for acquiring causality knowledge.

## 6 Conclusion

This paper presents a novel approach for modeling inter-topic preferences of users on Twitter. Designing high-quality linguistic patterns for locating support and opposition statements, we extracted users' preferences to various topics from a large collection of tweets. We formalized the task of modeling inter-topic preferences as a matrix factorization that maps both users and topics into a latent feature space that abstracts users' preferences. The experimental results demonstrate that this approach can predict missing topic preferences of users accurately (80–94%) and that the latent vector representations of topics encode inter-topic preferences.

An immediate future work is to embed the topic and user vectors to a cross-topic stance detector. It is possible to generalize this work for modeling heterogeneous signals such as interests and behavior of people, for example, "those who are interested in $A$ also support $B$," and "those who favor $A$ also vote for $B$". Thus, we believe that this work will bring a new application of NLP to other disciplines.

## Acknowledgements

(Removed for the reviewing process)

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
