# Peer review of "Other Topics You May Also Agree or Disagree: Modeling Inter-Topic Preferences using Tweets and Matrix Factorization"

_ACL 2017 — decision unknown_

[Official Review · Reviewer 1 · rating 3 · confidence 2]
soundness 3 · originality 4 · clarity 4 · impact 3 · substance 4 · appropriateness 5 · meaningful comparison 4 · presentation format Poster

- Strengths:

The deviation between "vocal" users and "average users" is an interesting
discovery that could be applied as a way to identify different types of users.

- Weaknesses:

I see it as an initial work on a new topic that should be expanded in the
future. A possible comparison between matrix factorization and similar topics 
in distributional semantics (e.g. latent semantic analysis) would be useful. 

- General Discussion:

In this paper, the authors describe an approach for modeling the
stance/sentiment of Twitter users about topics. In particular, they address the
task of inter-topic preferences modeling. This task consists of measuring the
degree to which the stances about different topics are mutually related.This
work is claimed to advance state of the art in this task, since previous works
were case studies, while the proposed one is about unlimited topics on
real-world data.The adopted approach consists of the following steps: A set of
linguistic patterns was manually created and, through them, a large number of
tweets expressing stance towards various topics was collected. Next, the texts
were expressed as triples containing user, topic, and evaluation. The
relationships represented by the tuples were arranged as a sparse matrix. After
matrix factorization, a low-rank approximation was performed. The optimal rank
was identified as 100. The definition of cosine similarity is used to measure
the similarity between topics and, thus, detect latent preferences not
represented in the original sparse matrix. Finally, cosine similarity is also
used to detect inter-topic preferences.A preliminary empirical evaluation shows
that the model predicts missing topics preferences. Moreover, predicted
inter-topic preferences moderately correlate with the corresponding values from
a crowdsourced gold-standard collection of preferences. 
According to the overview discussed in the related work section, there are no
previous systems to be compared in the latter task (i.e. prediction of
inter-topic preferences) and, for this reason, it is promising.

I listed some specific comments below.

- Rows 23 and 744, "high-quality": What makes them high-quality? If not
properly defined, I would remove all the occurrences of "high-quality" in the
paper.

- Row 181 and caption of Figure 1: I would remove the term "generic."

- Row 217, "This section collect": -> "We collected" or "This section explains
how we collected"- Row 246: "ironies" -> "irony"

- Row 269, "I support TPP": Since the procedure can detect various patterns
such as "to A" or "this is A," maybe the author should explain that all
possible patterns containing the topic are collected, and next manually
filtered?

- Rows 275 and 280, "unuseful": -> useless

- Row 306, "including": -> are including

- Row 309:  "of" or "it" are not topics but, I guess, terms retrieved by
mistakes as topics. 

- Rows 317-319: I would remove the first sentence and start with "Twitter
user..."

- Rows 419-439: "I like the procedure used to find the optimal k. In previous
works, this number is often assumed, while it is useful to find it
empirically."

- Row 446, "let": Is it "call"?